# Chronic Pelvic Pain in Congestion Pelvic Syndrome: Clinical Impact and Electromyography Pelvic Floor Activity Prior to and after Endovascular Treatment

**DOI:** 10.3390/jpm14060661

**Published:** 2024-06-20

**Authors:** Fabio Corvino, Francesco Giurazza, Milena Coppola, Antonio Tomasello, Francesco Coletta, Crescenzo Sala, Romolo Villani, Bernardo Maria de Martino, Antonio Corvino, Raffaella Niola

**Affiliations:** 1Section of Radiology, Department of Biomedicine, Neuroscience and Advanced Diagnostics (BiND), University Hospital “Paolo Giaccone”, 90127 Palermo, Italy; 2Interventional Radiology Department, A.O.R.N. “A. Cardarelli”, 80131 Naples, Italy; francesco.giurazza@aocardarelli.it (F.G.); milena.coppola@aocardarelli.it (M.C.); raffaella.niola@aocardarelli.it (R.N.); 3Emergency and Acceptance Department, Anesthesia, Emergency and Burn Intensive Care Unit, A.O.R.N. “A. Cardarelli”, 80131 Naples, Italy; antonio.tomasello@aocardarelli.it (A.T.); francesco.coletta@aocardarelli.it (F.C.); crescenzo.sala@aocardarelli.it (C.S.); romolo.vilanni@aocardarelli.it (R.V.); 4Clinical Neurophysiology Unit, A.O.R.N. “A. Cardarelli”, 80131 Naples, Italy; 5Medical, Movement and Wellbeing Sciences Department, University of Naples “Parthenope”, 80133 Naples, Italy; antonio.corvino@uniparthenope.it

**Keywords:** sclero-embolization, chronic pelvic pain, congestion pelvic syndrome, electromyography, pelvic floor activity

## Abstract

Background: This study aims to characterize the clinical impact of endovascular treatment in Chronic Pelvic Pain (CPP) patients due to Pelvic Congestion Syndrome (PCS) and to assess the diagnostic value of surface electromyography (sEMG) studies of pelvic floor musculature (PFM) in PCS patients pre- and post-endovascular treatment. Between January 2019 and July 2023, we studied consecutive patients who were referred for interventional radiology assessment and treatment to a tertiary trauma care hospital, had evidence of non-obstructive PCS from Magnetic Resonance Imaging (MRI), had sEMG of PFM and who had undergone endovascular treatment. The primary outcome was clinical, defined as a change in symptom severity after endovascular treatment. The secondary outcome was a difference in the sEMG values pre- and post-endovascular therapy. Results: We included 32 women (mean age 38 years). CPP was the leading symptom in 100% patients, followed by dysmenorrhea (75%) and post-coital pain (68.7%). Endovascular therapy included ovarian vein embolization in 28 patients (87.5%) and internal iliac vein embolization in only 2 patients (6.2%). After a median of 8 (range 6–10) months from endovascular treatment, 29 (90%) of patients reported an improvement of the main symptoms, and 15 (46%) were symptom-free. The sEMG values did not show a statistical difference pre- and post-PCS endovascular treatment. Conclusions: Endovascular treatment appeared to be highly effective in CPP due to PCS and was associated with a low rate of complication. sEMG study could be useful in revealing alterations of PFM electrophysiology, but a difference pre- and post-embolization in PCS patients was not demonstrated.

## 1. Introduction

Chronic pelvic pain (CPP) is a complex and disabling clinical condition among women of childbearing age; about one quarter of women of reproductive age complain of this clinical condition lasting over one year in duration. The clinical symptoms are generally vague and inconsistent, reflecting the multiple factors that have a role in its pathogenesis. The American College of Obstetricians and Gynecologists defines CPP as “pain of the pelvic region, sensed from the pelvic organs themselves that typically lasts more than 6 months”. Musculoskeletal, neurological and psychological conditions have been implicated in presentations of pelvic pain, but no diagnosis is made in over 60% of cases [1].

In 1948, Taylor first proposed a connection between the presence of pelvic varicosities and CPP [2]; this condition was classified as a clinical entity, known as pelvic congestion syndrome (PCS), only after Beard et al.’s work of the 1980s, in which more than 90% of women with CPP presented with associated pelvic varicosities [3]. However, more recent reports described a prevalence of 39% of women with CPP and about 30% of these patients have a pelvic venous insufficiency due to PCS [4,5].

PCS is one of the main causes of CPP, accounting for about 30% of patients who have a pelvic venous insufficiency. The pathophysiology of PCS pain includes an array of visceral, somatic and neurological pain generators [6]. In chronic lower extremity venous insufficiency, the pain state is related to abnormal vessel function [7]; about four decades ago, Kaupilla et al. demonstrated a higher rate of histological vascular anomalies (such as fibrosis, muscular hypertrophy and widened ovarian diameter) in a group of PCS patients affected by CPP [8]. However, more recent views of pain states emphasize both peripheral and central contributors to nociception. Indeed, in some males affected by chronic pelvic pain, the presence of periprostatic pelvic varices have been described, suggesting a relationship between the autonomic nervous system dysregulation of venous tone and some pelvic pain states [9]; similarly, the cerebral vasodilation seen in migraine headaches is thought to reflect neurological changes of the trigeminal system [10].

A correlation between CPP and dysfunction of the pelvic floor musculature (PFM) has been demonstrated; surface electromyography (sEMG) seems to be a useful tool to evaluate these kinds of dysfunctions.

CPP is closely associated with dysfunction of the PFM, and the sEMG turned out to be a useful tool to assess these kinds of patients. sEMG is the electric signal recorded by means of electrodes attached to the surface of the skin produced by the sum of the extracellular potential from the active muscle fibers beneath the electrodes. A greater power of PFM activity has been demonstrated in CPP patients than in healthy women, especially when compared with mature/parous subjects; this difference is related to an increased number of motor units recruited and the muscle fatigue caused by overactivation of the PFM [11,12].

The aim of our retrospective observational study is to analyze the clinical impact of endovascular treatment in CPP patients due to PCS and to assess the diagnostic value of the sEMG study of PFM in PCS patients pre- and post-endovascular treatment.

## 2. Materials and Methods

*Patients.* Between January 2019 and July 2023, 32 consecutive patients (mean age, 38 years; 29–60) with a clinical diagnosis of CPP due to type 1 Greiner’s classification PCS underwent endovascular treatment in a major hospital in an Italian metropolitan city [13]. An Excel database was prospectively recorded and retrospectively analyzed. Data were collected from medical reports, discharge letters and endovascular procedure reports, including demographic information, relevant comorbidities and symptoms. Moreover, a clinical score, derived from a previous manuscript, was used to diagnose PCS and to evaluate clinical improvement [14]. The main demographic and clinical characteristics of patients are shown in Table 1.

Overall, 26 (81.2%) of 32 patients had had previous pregnancies (mean number of deliveries: 2; 1–3) and only 2 had a previous history of endometriosis, treated with laparoscopic surgery in one case and with hormones in both cases. CPS syndrome was diagnosed by a gynecologist with specific experience in this pathology, both on the basis of symptoms and on transvaginal ultrasound (TVU) examination. The most prevalent symptoms were CPP (100% of patients), dysmenorrhea (75%), post-coital pain (68.7%), dyspareunia (50%) and leg varicosities (48.8%) (Table 2). In the presence of a clinical suspicion of CPP related to PCS, the patients underwent a TVU study. The ultrasound parameters used to diagnosis PCS were the presence of a pelvic vein or venous plexus equal to or greater than 8 mm; reversed or altered flow during Valsalva; crossing veins in the myometrium and an association with polycystic changes of the ovaries. Pre-procedural Magnetic Resonance Imaging (MRI) was performed in all patients to exclude the presence of PCS related to nutcracker syndrome, May–Thurner, or one of the other CPP secondary causes (endometriosis, fibroids, pelvic inflammatory disease, adenomyosis, irritable bowel syndrome, pelvic floor pain, ovarian cancer and others).

The inclusion criterion was the presence of non-cyclical pelvic pain for at least 6 months associated with TVU or MRI findings compatible with PCS syndrome. Exclusion criteria to treatment included all the contraindications to endovascular procedure, including active pelvic infection, severe contrast medium allergy, coagulopathy, the presence of any active malignancy or history of it in the previous one year, previous hysterectomy or oophorectomy and pregnancy.

This study was approved by the local ethics committee and was conducted according to the principles of the Declaration of Helsinki. The human ethics review board at our institution approved the study design. Informed consent for data collection was obtained from all patients. To characterize PFM activity, a sEMG was carried out at baseline before endovascular treatment and 6 months after it. The latency of the P40 response (n.v. 37.68 ± 2.60 ms), the P1-N1 amplitude (n.v. 119 3.64 ± 1.01 μV) and the latency of the R1 early response (n.v. < 45 ms) were evaluated in the paraclitoral site (Figure 1).

*Endovascular technique.* The procedure was undertaken in inpatient conditions with mild analgosedation. Bilateral ovarian and iliac diagnostic subtraction phlebography was performed from femoral access to delineate the anatomy and identify the main vessels and collateral pathway, obtained with and without provocation such as the Valsalva maneuver. Ovarian catheterization was achieved with a 5 Fr HET or another shaped-tip catheter (COOK Medical, Bloomington, IN, USA) with a 0.038″ lumen depending on the type of coil utilized. In the case where the 0.018″ micro-coils platform was utilized, a 2.7 Fr Progreat Terumo (Terumo Medical, Tokyo, Japan) was used for delivery. A sandwich technique was preferred in all cases with pushable coils used after the distal injection of 3% sodium tetradecyl sulphate (STS), prepared as a foam according to the Tessari method, with a maximum dose per procedure of 10 mL [15]. Complete occlusion with the entire length of the incompetent vein prevented possible future recanalization. Coils’ common sizes range from 8 mm to 20 mm, and are generally oversized to prevent migration. Internal iliac veins were embolized with the same sandwich technique in cases where a venous reflux was confirmed at phlebography during the Valsalva maneuver. Final phlebography control after embolization was undertaken to confirm vessel occlusion. In patients with symptomatic uterine myomas, a bilateral uterine artery embolization (UAE) was performed in the same session. All procedures were performed by an interventional radiology radiologist with more than 5 years of experience. The primary cause of PCS was left ovarian vein reflux, highlighted in all 32 patients (100%) (Figure 2), while in 8 cases the insufficiency was bilateral; moreover, in 2 cases internal iliac vein insufficiency was demonstrated. Ovarian vein embolization was performed in all 32 patients (100%) after demonstrating reflux during the phlebographic study (just on the left side in 24 patients (75%) and bilaterally in 8 patients (25%)). The internal iliac veins were embolized only in 2 cases (6.25%), 1 on the right side and 1 on the left side, where the reflux was demonstrated from leakage points after bilateral ovarian embolization. Findings from pelvic phlebography with fluoroscopy and therapeutic details are summarized in Table 3.

*sEMG signal Recording Technique.* The sEMG signal was recorded in each patient from the left and right sides of the pelvic floor in a dorsal lithotomy position with adhesive electrodes. The study of the sacral evoked potentials was carried out with a manual bipolar stimulator with a cathode at the paraclitoral site and recorded using surface electrodes in the parietal region CZ (−2 cm)—FPZ according to the international 10–20 system, with square wave stimulation of the duration of 0.5 ms, intensity 3–4 times the sensory threshold, frequency 1.5 Hz, 151 impedance < 5 kOhm, bandwidth 1 Hz–3 kHz, 200 stimuli. The latency of the P40 response (n.v. 37.68 ± 2.60 ms) and the P1-N1 amplitude (n.v. 3.64 ± 1.01 μV) were evaluated. The anal pudendal reflex was evaluated with a manual bipolar stimulator with the cathode positioned in the paraclitoral site and recorded by means of coaxial needle electrodes from the external anal sphincter muscle. The latency of the R1 early response was evaluated (n.v. < 45 ms).

*Study endpoints and definition.* Primary clinical success was defined as a change in symptoms severity following 6 months of endovascular therapy. Symptoms relief was divided into three categories relating to its improvement: 1. complete freedom from PCS symptoms; 2. improvement with residual symptoms; and 3. no improvement or changes in symptoms.

Secondary clinical outcome was defined as a difference in the sEMG study between the single parameters studied before and after endovascular treatment.

Technical success was defined as a successful endovascular procedure with no further evidence at imaging of reflux.

Safety outcome was defined as the prevalence of peri- and post-operative complications, categorized according to CIRSE guidelines [16].

*Follow-up and clinical outcome.* The median follow-up was 8 months (range: 6–10 months). Clinical improvement was demonstrated in 29 patients (90%—Category 2), with 15 patients (46%—Category 1) having a complete resolution of symptoms. In three patients (9.3%—Category 3), endovascular treatment did not improve symptoms. Of these three patients, one had a history of endometriosis with previous multiple surgical laparoscopic interventions, and the second one had a previous history of sacral trauma associated with a significant increase in R1 latency of the anal pudendal reflex, directly related to a neuronal impairment. The last one had a previous history of urolithiasis with a different previous urological procedure. In one case with no resolution of vulvar varices after ovarian sclero-embolization, the patient underwent direct percutaneous STS foam sclerotherapy with complete resolution. The latency values of P40 response pre-treatment are slightly reduced compared to normal values; compared to the pre-treatment value, the latency of P40 response was slightly increased after treatment but not with a statistical significance. An increase in P1-N1 signal amplitude in PCS was demonstrated; moreover, a reduction in the signal amplitude after PCS treatment was demonstrated but with no statistical significance. There was no statistically significant difference before and after endovascular treatment in the study of R1 latency of the anal pudendal reflex. The sEMG values pre- and post-embolization are summarized in Table 4. Complications due to endovascular procedure occurred in four patients (12.5%): in two patients (6.25%), partial coil migration into the inferior vena cava in one patient and into the left renal vein in another one was promptly retrieved by snaring techniques during endovascular procedure (Grade 1 CIRSE classification). Two patients (6.25%) suffered mild symptoms of pelvic thrombophlebitis, successfully treated with low-molecular-weight heparin (LMWH) (Grade 3 CIRSE Classification).

## 3. Discussion

CPP has a worldwide population prevalence of 15% in the female population between 18 and 50 years, and in 61% of cases the etiology is unexplained. Labeling pelvic pain patients with single pathologic diagnoses such as PCS may hinder treatment efforts; however, a prevalence estimate of around 30% of PCS in CPP patients has been demonstrated [17].

The pathogenesis of PCS is still unclear, but it has been demonstrated to be multifactorial in nature. Pregnancy hyper-flow venous condition, in terms of gonadal and pelvic plexus dilation, due to estrogen level vasodilator effects, could be one of the pathogenetic factors involved. A main role, in addition, is related to a valve incompetence development and/or stenosis/obstruction onset of the draining veins [18]. The type 1 Greiner’s classification PCS treated in our study, that is, a non-obstructive left ovarian vein insufficiency and reflux, is the most common pathophysiological cause, with a prevalence of more than 50% of cases. In 2021, Meissner et al. proposed the Symptoms–Varices–Pathophysiology (SVP) classification of pelvic venous disorders; however, its effective role in determining treatment is still to be demonstrated [19]. A novel PCS management strategy proposed an association between a simple screening tool—the PCS score, which gynecologists and primary care physicians could use—and accurate noninvasive imaging; this synergy, also used in our study, seems to improve the PCS diagnosis and successful treatment rate [14].

A non-invasive imaging study is mandatory in PCS management strategy, primarily to confirm the clinical suspicion of PCS and secondly to exclude other CPP causes [17]. Ultrasound (US), both transabdominal and transvaginal, with color Doppler imaging and Doppler spectral analysis, should be used in PCS diagnosis. A dilated and tortuous pelvic vein (diameter > 6 mm), a slow blood flow (<3 cm/s) or reversal caudal flow, a dilated arcuate veins and polycystic changes in the ovaries are the ultrasound diagnostic criteria for PCS diagnosis [20]. MRI represents the first line of investigation, providing better imaging of the many causes of CPP and, due to the patients’ young ages, should be preferred to Computed Tomography (CT) given the absence of ionizing radiation. MRI has shown a sensitivity and specificity of 88% and 67%, respectively, in demonstrations of ovarian vein congestion. However, CT and MRI studies are both performed in the supine position, thus allowing an underestimation of venous dilation compared to US and phlebography, where provocative maneuvers could be employed. In our cases, a prior MRI study with a demonstration of pelvic venous congestion for ovarian veins and MR angiography sequence with the detection of vein dilatation and reflux was mandatory [21]. Catheter phlebography still remains the gold standard for PCS diagnosis, but its diagnostic role is reserved only for cases when noninvasive studies are inconclusive [22].

PCS treatments that have been proposed and used over time are multiple and include conservative, psychological, medical, surgical and endovascular options, both in isolation and in combination. An RCT comparing combined hormone suppression and psychotherapy experienced sustained CPP relief with a >50% reduction 262 in pain score [23]. A medical therapy, alone or in combination with other treatment, did not demonstrate a superior efficacy to other treatment but only an increase in adverse symptoms if used in a long-term setting [24]. Surgical treatments include different alternatives, including a mini-invasive laparoscopic approach for ovarian vein ligation to more invasive ones, such as hysterectomy and/or oophorectomy. Only one study, published by Gargiulo et al., demonstrated an extraordinary 100% symptom relief in 23 women treated with a laparoscopic ovarian vein ligation treatment [25]. Finally, hysterectomy, proposed when all other treatments have failed, showed a 22–33% failure rate in clinical symptom resolution [26].

The endovascular management of PCS is recommended with level 2B evidence by the Society of Vascular Surgery: “We suggest treatment of pelvic congestion syndrome and pelvic varices with coil embolization, plugs, or trans-catheter sclerotherapy, used alone or together” [27]. Our study confirms the actual literature about the difference rate between technical success (almost always described at above 95%) and clinical success (usually between 58% and 100%), regardless of the embolic agent. To date, there have not been any randomized studies comparing clinical success using different embolic agents for PCS endovascular treatment. The choice of embolic agent is generally operator-dependent, with no different outcomes described with the use of a specific agent [17]. Moreover, such a high technical success with a low complication rate allows us to conclude that endovascular techniques are safe and effective in PCS syndrome treatment.

The clinical outcomes are in line with the previous literature and could be considered as overall positive: 90% of patients showed a clinical improvement, with 46% of patients having a complete resolution of symptoms; in only three patients (9.3%) did endovascular treatment not improve symptoms. A recent meta-analysis demonstrated a short-term outcome similar to our study, showing a clinical improvement in 88% of patients and 6–32% of patients who did not experience any symptom relief [28]. Another comprehensive review in 2018 stated that of the 1308 patients included in the study, about 75% reported early symptom relief after embolization [29].

CPP and other clinical conditions of the pelvic floor (i.e., fecal incontinence, organ prolapse, etc.) are closely associated with the disfunction of the pelvic floor musculature PFM, so sEMG could be a useful tool to assess patients’ clinical conditions [11]. In our study, the baseline value of the latency of p40 response and P1-N1 signal amplitude confirm that a slight muscular hypertonia is associated with CPP due to PCS; however, these values after PCS endovascular treatment indicate that this correlation is not directly connected with venous insufficiency, but there are subtle other pathologies related that should be studied further. The R1 latency of the anal reflex comparison showed no difference between pre- and post-treatment; this type of evaluation is directly related to neuronal impairment, which was highlighted in only one patient. The results obtained prove that sEMG can reveal alterations in the PFM electrophysiology associated with CPP related to PCS and provide clinicians with objective information that can help them to better evaluate CPP patients’ conditions, thus allowing more efficient management of this complex syndrome. Future studies of PCS-related pelvic pain should also evaluate central mediators of vascular tone, perhaps by employing biofeedback or stress reduction. Anxiety and depression in this group of women has long been recognized, with much debate as to whether this is cause or effect. Farquhar et al. postulated that in a population of PCS patients, only women on combined hormone suppression and psychotherapy experienced sustained CPP relief, pointing out that overlapping neurological and hormonal factors can influence pain symptom expression [23].

Even though the sample size was homogenous, our study has several limitations. First, the sample size studied was small. Second, the evaluation of only patients with a type 1 Greiner’s classification PCS makes the population more homogeneous, ruling out patients with type 2 and 3 of PCS syndrome, which accounts for more than 40% of patients with PCS. Third, all patients were recruited by only one interventional radiology center, which could bring about a selection bias. Last, there was an the absence of a comparison healthy population studied with sEMG. However, given the absence in the literature of randomized controlled trials on management and treatment strategies of PCS, also with sEMG studies of PFM, our data may provide a reference for a future trial.

## 4. Conclusions

Currently, the management of women with chronic pelvic pain is suboptimal, with only 40% of affected women being referred to a dedicated specialist. Endovascular therapy should be chosen over a less effective medical approach (psychotropic, hormonal or venoactive drugs) or a more invasive surgical approach, highlighting that PCS is a multifactorial disease in which vein disease is only one side of a more complex pathology [4]. The slight differences found between PFM activity in PCS patients before and after endovascular treatment prove that sEMG could be a useful tool for selected patients, and more studies are necessary to identify its role in CPP clinical diagnosis and management.

## Figures and Tables

**Figure 1 jpm-14-00661-f001:**
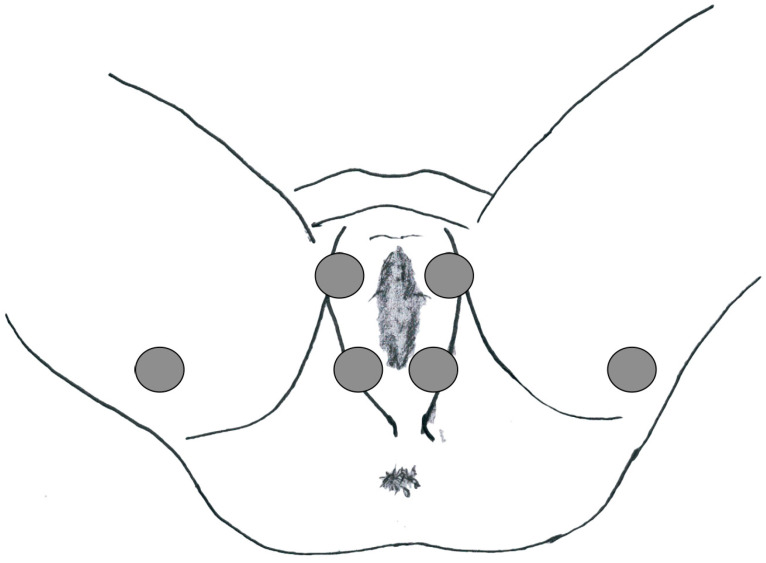
Electrode location for the PFM sEMG study.

**Figure 2 jpm-14-00661-f002:**
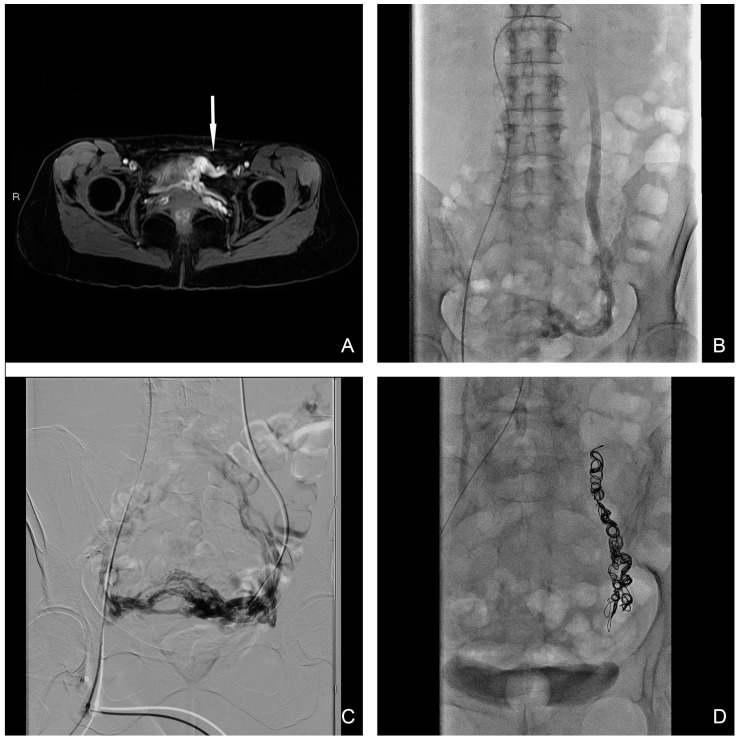
(**A**) T1 water–lava axial MRI scan shows peri-uterine plexus bilaterally dilatated with tortuous course and a maximum caliber of 6 mm (white-arrow). (**B**,**C**) Selective and superselective diagnostic phlebography through a 5 Fr catheter demonstrates ovarian vein dilation with reflux in the para-uterine veins and partial drainage in internal iliac veins. (**D**) Final image after sclero-embolization procedure.

**Table 1 jpm-14-00661-t001:** Main demographic and clinical characteristics of patients.

Patients	32
Age (years), mean ± SD	38.0 ± 5.4
BMI, mean ± SD	21.8 ± 2.8
Number of pregnancies, mean ± SD	1.6 ± 0.8
Comorbidities	
Uterine myomas, n (%)	5 (15.6)
Endometriosis, n (%)	2 (6.25)
Varicose vein, n (%)	18 (56.25)
Pelvic cysts, n (%)	4 (12.5)
Deep vein thrombosis, n (%)	2 (6.25)
Acquisition of Arteriovenous Malformation, n (%)	1 (3.1)

**Table 2 jpm-14-00661-t002:** Symptoms and PCS scores at baseline of the study population.

Symptoms	Baseline
Chronic lower abdominal pain, n (%)	32 (100)
Dysmenorrhea, n (%)	24 (75)
Postcoital pain, n (%)	22 (68.75)
Leg varicosities, n (%)	15 (48.8)
Vulvar varicosities, n (%)	3 (9.3)
Orthostatic pain, n (%)	10 (31.25)
Dyspareunia, n (%)	16 (50)
Back pain, n (%)	8 (25)
Hemorrhoids, n (%)	4 (12.5)
Depression, n (%)	2 (6.25)
Dysuria, n (%)	2 (6.25)
PCS score (mean ± Sd)	7.2 ± 1.9

**Table 3 jpm-14-00661-t003:** Diagnostic and procedural data of patients.

Main Pathology	
Ovarian vein insufficiency, n (%)	32 (100)
Left, n (%)	32 (100)
Right, n (%)	8 (25)
Internal iliac vein insufficiency, n (%)	2 (6.25)
**Phlebographic** **findings**	
Ovarian vein reflux, n (%)	32 (100)
Internal iliac vein reflux, n (%)	2 (6.25)
Vulvar vein reflux	1 (3.1)
**Treatment**	
Ovarian vein embolization, n (%)	32 (100)
Left, n (%)	24 (75)
Bilateral, n (%)	8 (25)
Internal iliac vein embolization, n (%)	2 (6.25)
Left, n (%)	1 (3.25)
Right, n (%)	1 (3.25)

**Table 4 jpm-14-00661-t004:** The sEMG values evaluated pre- and post-PCS sclero-embolization.

sEMG Signals	Pre-Embolization	After Embolization	*p* Value
Latency p40 response	32.85 ± 1.40 ms	34.05 ± 2.30 ms	≥0.05
P1-N1 amplitude	4.26 ± 0.96 µV	3.96 ± 0.85 µV	≥0.05
Anal pudendal reflex	40 ± 2 ms	39 ± 3 ms	≥0.05

## Data Availability

The raw data supporting the conclusions of this article will be made available by the authors on request.

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
