# Peer review of "Chronic Pelvic Pain in Congestion Pelvic Syndrome: Clinical Impact and Electromyography Pelvic Floor Activity Prior to and after Endovascular Treatment"

_jpm, 2024, doi:10.3390/jpm14060661_

Round 1

Reviewer 1 Report

Comments and Suggestions for Authors

Dear Authors, I find your research both original and interesting! Nevertheless, the way in which it is narrated is not at all reader-friendly, as the findings are presented in a rather dull manner. Another  minus is represented by the fact that the manuscript needs extensive English language corrections. Moreover, in my opinion, the discussion section could be more compact. I would comprise the information in the entire article, as many ideas are repeated. Respectfully yours, a Reviewer.

Comments on the Quality of English Language

Very bad. Major errors. Bad English language totally affects the quality of the (scientific) presentation. 

Author Response

Response to Reviewer X Comments

1. Summary

Thank you very much for taking the time to review this manuscript. Please find the detailed responses below and the corrections highlighted in the re-submitted files.

2. Questions for General Evaluation

Reviewer’s Evaluation

Response and Revisions

Does the introduction provide sufficient background and include all relevant references?

Can be improved

Are all the cited references relevant to the research?

Can be improved

Is the research design appropriate?

Can be improved

Are the methods adequately described?

Must be improved

We extensively review the Material and Methods section in which we add more information about the patients’ selection (line 100 to 113).

Are the results clearly presented?

Can be improved

Are the conclusions supported by the results?

Can be improved

3. Point-by-point response to Comments and Suggestions for Authors

Comments 1: Dear Authors, I find your research both original and interesting! Nevertheless, the way in which it is narrated is not at all reader-friendly, as the findings are presented in a rather dull manner. Another minus is represented by the fact that the manuscript needs extensive English language corrections. Moreover, in my opinion, the discussion section could be more compact. I would comprise the information in the entire article, as many ideas are repeated. Respectfully yours, a Reviewer.

Response 1: Agree with you; for this reason, our fellow native speaker extensively review the manuscript. These are the changes:

Line 42 Page 1: We add of in the sentence complain of this clinical condition

Line 45 Page 2: we correct Gynecologists

Line 46 Page 2: we correct the sentence “typically lasts more than 6 months”

Line 52 Page 2: we correct the sentence “presented with associated pelvic varicosities”

Line 53 Page 2: we correct the word “reports”

Line 54 Page 2: we correct the sentence “due to PCS”

Line 55 Page 2: we correct the sentence “accounting for about 30% of these patients”

Line 84 Page 2: we correct the word “of” in “in”.

Line 115 Page 3: we correct the word “ethic” in “ethics”.

Line 118 Page 3: we correct the word “prior” in “before”.

Line 129 page 4: we correct the word “due to” in “depending”.

Line 131 Page 4: we correct the words “to delivery” in “to deliver”.

Line 140 Page 4: we correct the sentence “was carried out” in was performed”.

Line 145-147 Page 4: we correct the sentence “after demonstrating reflux during the phlebographic study: just on the left side in 24 pa-tients (75%) and bilaterally in 8 patients (25%)”

Line 156 Page 5: we correct the word “in” in “to”.

Line 204 Page 7: we add a comma

Line 214 Page 7: we correct the sentence “could one of the pathogenetic factor involved” in “could be one of the pathogenetic factors involved”.

Line 236 Page 7: we add a comma

Line 308 Page 9: we correct the word “prior” in “before”

Line 309 Page 9: we deleted a before “selected patients”

4. Response to Comments on the Quality of English Language

Point 1: Very bad. Major errors. Bad English language totally affects the quality of the (scientific) presentation.

Response 1: Agree with you. See the previous answer.

Reviewer 2 Report

Comments and Suggestions for Authors

Excellent article, well written and very interesting.

Line 83

Only MRI was used to diagnose CPP (Greiner type 1). It's a shame not to have also used ultrasound scores through a transvaginal approach...

I would like it to be better specified the difference between technical and clinical success.

Why for example

between lines 243-245 we talk about

“difference between the technical success rate (almost always described above 95%) and the clinical success (usually between 58% and 100%), regardless of the embolic agent”

while in lines 252-254 it says

“90% of patients showed clinical improvement and 46% of patients had complete resolution of symptoms; only in 3 patients (9.3%) the endovascular treatment did not improve the symptoms"...

Author Response

Response to Reviewer X Comments

1. Summary

Dear Editor,

first of all, we would like to thank you and reviewers for your time invested to provide comments to our manuscript.

We have read carefully reviewers’ observations and we have answered point by point.

Please find in the pages below our answers.

We hope we have improved the manuscript according to their revisions.

We really care about the possibility to publish our experience on your journal and we hope “Journal Personalized Medicine ” readers could appreciate our work.

Our bests,

The Authors

2. Questions for General Evaluation

Reviewer’s Evaluation

Response and Revisions

Does the introduction provide sufficient background and include all relevant references?

Yes

Are all the cited references relevant to the research?

Yes

Is the research design appropriate?

Yes

Are the methods adequately described?

Can be improved

Are the results clearly presented?

Can be improved

Are the conclusions supported by the results?

Can be improved

3. Point-by-point response to Comments and Suggestions for Authors

Comments 1: Excellent article, well written and very interesting.

Response 1: Thank you for this statement. Our work was very hard to explore the universe of CPS patients.

Comments 2: Line 83

Only MRI was used to diagnose CPP (Greiner type 1). It's a shame not to have also used ultrasound scores through a transvaginal approach...

Response 2: Agree. However, the MRI study is necessary to exclude the presence of other causes of Chronic Pelvic Pain (CPP) because in out study we consider patient with CPS diagnosis. We add the role of MRI as, beyond that to exclude the presence of a PCS related to a nutcracker or a May-Thurner syndrome also for other CPP secondary causes as Endometriosis, Fibroids, Pelvic inflammatory disease, Adenomyosis, Irritable bowel syndrome, Pelvic floor pain, Ovarian cancer and others). We add a specific sentence where we describe the diagnosed by whom it was carried out. Moreover, we have also catalogued the specific TVG findings that allow us to diagnose PCS.

All the changes are in bold type (Line 94-106).

Comments 3: I would like it to be better specified the difference between technical and clinical success.

Why for example

between lines 243-245 we talk about

“difference between the technical success rate (almost always described above 95%) and the clinical success (usually between 58% and 100%), regardless of the embolic agent”

while in lines 252-254 it says

“90% of patients showed clinical improvement and 46% of patients had complete resolution of symptoms; only in 3 patients (9.3%) the endovascular treatment did not improve the symptoms"...

Response 2:  Agree. Because this is the main issue of the PCS treatment. In the majority of the manuscript present in the literature the technical success is always high (generally up to 95%); however, in more than 40%-50% of cases there isn’t a complete resolution of the symptoms. For this reason, we try to demonstrate the presence of a muscular hypertonicity of the pelvic floor musculature.

In our specific case we considered three different changes in symptoms severity (Line 160-163) ad to better understand the results we add in follow-up and clinical outcome section the category after the percentage.

4. Response to Comments on the Quality of English Language

Point 1: None

Response 1:  None

5. Additional clarifications

Reviewer 3 Report

Comments and Suggestions for Authors

Thank you very much for your well written paper. I would only suggest changing "Informed consent for data collection was obtained from all conscious patients" - to all patients, deleting conscious. It does not look like you had any unconscious patients, did you?

For patients with symptomatic leiomyoma, I wonder what were the symptoms and if they have been treated for their symptoms apart from their CCP? Please add.

Please add your inclusion and exclusion criterias (for example - any malignancies, who did not show up for the 6 month follow up etc) and the numbers you have begin and ended with

Author Response

Response to Reviewer X Comments

1. Summary

Dear Editor,

first of all, we would like to thank you and reviewers for your time invested to provide comments to our manuscript.

We have read carefully reviewers’ observations and we have answered point by point.

Please find in the pages below our answers.

We hope we have improved the manuscript according to their revisions.

We really care about the possibility to publish our experience on your journal and we hope “Journal Personalized Medicine ” readers could appreciate our work.

Our bests,

The Authors

2. Questions for General Evaluation

Reviewer’s Evaluation

Response and Revisions

Does the introduction provide sufficient background and include all relevant references?

Yes

Are all the cited references relevant to the research?

Yes

Is the research design appropriate?

Yes

Are the methods adequately described?

Yes

Are the results clearly presented?

Yes

Are the conclusions supported by the results?

Yes

3. Point-by-point response to Comments and Suggestions for Authors

Comments 1: Thank you very much for your well written paper. I would only suggest changing "Informed consent for data collection was obtained from all conscious patients" - to all patients, deleting conscious. It does not look like you had any unconscious patients, did you?

Response 1: Thank you for this statement. We’ll delete it.

Comments 2: For patients with symptomatic leiomyoma, I wonder what were the symptoms and if they have been treated for their symptoms apart from their CCP? Please add.

Response 2: Agree. However, as the nature of CPP is not well defined in case of patients where we demonstrated both the CPS and leiomyomas, we preferred to treat both in the same session. In our population the more common symptom of leiomyomas was CPP.

Comments 3: Please add your inclusion and exclusion criterias (for example - any malignancies, who did not show up for the 6 month follow up etc) and the numbers you have begin and ended with

Response 3: We added the exclusion criteria from line 107 to 112. The number of patients that we studied was the same from the begin because we don’t find any of the exclusion criteria in all of them.
